# OpenReview forum: "AdaCoT: Pareto-Optimal Adaptive Chain-of-Thought Triggering via Reinforcement Learning"
_ICLR.cc/2026/Conference — Submitted to ICLR 2026_

### Official Review · Reviewer_moGN · 2025-10-26

**Soundness:** 2
**Presentation:** 2
**Contribution:** 2
**Rating:** 2
**Confidence:** 4

**Summary:**

This paper addresses the significant computational overhead of CoT, which is often wastefully applied to simple queries that do not require complex reasoning. The authors propose AdaCoT, a framework that enables LLMs to adaptively decide whether to invoke CoT for a given query. The core idea is to frame this adaptive decision as a Pareto optimization problem, seeking to find an optimal balance between two competing objectives: maximizing model performance (accuracy) and minimizing reasoning cost (CoT usage). This trade-off is managed via PPO where the reward function includes penalty coefficients ($\alpha_1$ for missing CoT, $\alpha_2$ for overusing CoT). By varying these coefficients, the authors can train a spectrum of models that trace the Pareto frontier.

A key technical challenge identified is "decision boundary collapse," where the adaptive policy reverts to always (or never) using CoT when trained on domain-specific data (e.g., math problems). To solve this, the authors introduce Selective Loss Masking (SLM), a technique that preserves the adaptive decision-making capability. Experiments show that AdaCoT models can define a superior Pareto frontier compared to "Full CoT" or "No CoT" baselines, achieving, for instance, 62.8% average score with only a 53.3% CoT rate, approaching the 65.0% score of the "Full CoT" model.

**Strengths:**

- Framing the adaptive reasoning problem as a Pareto optimization task is an elegant and principled approach. It moves beyond simple heuristics and provides a formal framework for navigating the complex trade-off between performance and computational cost.

- The results are compelling from an efficiency standpoint. The framework demonstrates a clear ability to substantially reduce CoT usage while maintaining high performance. The production traffic results (Table 2), showing CoT rates as low as 3.18% and token reductions of 69%, are particularly strong evidence of the method's practical utility.

**Weaknesses:**

- The core of the method relies on a binary "CoT vs. No CoT" decision. This is a brutal simplification of the reasoning problem. The true spectrum of reasoning is continuous: simple queries need 0 steps, moderate queries need a few steps, and complex queries need many steps. The proposed framework cannot distinguish between a 3-step reasoning chain and a 30-step one; it only decides whether to generate anything in the <think> tags. This binary design is a fundamental flaw that the rest of the method must compensate for. The authors even acknowledge this limitation in Section 4.1.

- A direct consequence of the flawed binary design is the method's reliance on manually tuning the $\alpha_1$ and $\alpha_2$ penalty coefficients to trace the Pareto frontier. This is not a "learned" trade-off but a brute-force sweep. This makes the method impractical for real-world deployment. How would a practitioner set these $\alpha$ values for a new task or data distribution without re-running this entire, expensive experimental sweep? A more sophisticated design would ideally learn the optimal reasoning length, not require manual tuning of arbitrary penalties.

- SLM as a Symptomatic Fix (a "Patch"): The "decision boundary collapse" problem and the SLM solution feel like an engineering patch for a problem created by the initial design. The model should collapse to a 100% CoT rate on a math dataset, as this is the optimal policy for that data. The SLM mechanism essentially prevents the model from learning the correct policy for that domain in order to preserve the "adaptive" behavior from a previous training stage. This suggests the multi-stage RL pipeline and binary reward are not robust, requiring this extra mechanism to hold them together.

- **Crucial Missing Experimental Baselines**: This is the most significant weakness of the paper. The paper claims to present a "Pareto-Optimal" solution but fails to compare it against any other competing reasoning efficiency methods. The related work (Section 5) correctly identifies other SOTA approaches (e.g., pruning, summarization, RL with brevity rewards). However, the experiments only compare AdaCoT against its own non-adaptive (Full CoT, No CoT) and non-RL (Adaptive SFT) ablations. The Pareto frontier plot in Figure 1 is misleading because it's missing the most important data points: where would other methods from the literature fall on this graph? Without this comparison, it is impossible to know if AdaCoT is truly "Pareto-optimal" or if it is dominated by other, simpler efficiency techniques. The claim of "pioneering adaptive triggering" (Abstract) is too strong when the work is not benchmarked against its peers.

- The title is incorrect as default template title.

**Questions:**

- Is SLM truly necessary if the reward structure was more robust? What happens if you simply remove the $P_{over}$ penalty during the RL-Math stage? Wouldn't the model learn to use CoT (as required by the task) and then re-learn to be frugal when $P_{over}$ is re-introduced in the RL-General stage? SLM feels like it's fighting the RL agent's correct adaptation.
- How sensitive is the framework to the initial SFT data labeling (e.g., the 67% CoT-required ratio)? To deploy AdaCoT in a new, specialized domain (e.g., legal, medical), would a user need to re-create the entire SFT dataset and re-run the full 2-stage RL pipeline, including the expensive $\alpha$-coefficient sweep?

---

> ### Author Response · Authors · 2025-12-04
> **Response to The Official Review (1/2)**
>
> Dear Reviewer,
>
> We are sincerely grateful for your thorough and deeply critical review.
> Your critiques on the binary design, the role of SLM, and the missing baselines are all valid and significant. We hope our response below clarifies our design philosophy and the specific problems we aimed to solve.
>
> Response to Weaknesses
>
> Weakness 1 & 2: The binary "CoT vs. No CoT" decision is a "brutal simplification" and relies on manual tuning of α, not learning.
>
> Thank you for raising this fundamental critique. We agree that the full spectrum of reasoning is continuous, and our binary approach is a deliberate simplification. However, we respectfully argue that it is a pragmatic and impactful simplification that addresses a distinct and primary question in efficient reasoning.
>
> Solving a Different Problem: Our work does not aim to solve the problem of "how long should the reasoning chain be?". Instead, it tackles the preceding, arguably more common, question: "is reasoning necessary at all?". In practice, the largest efficiency gains come from entirely skipping unnecessary reasoning for simple queries, which form a vast majority of real-world traffic (as evidenced by our 3.18% CoT rate on production data). Our binary framework is purposefully designed to capture this "zero-step" vs "N-step" decision, which provides the bulk of the cost savings. We see this as complementary to, not a replacement for, work on optimizing reasoning length.
>
> α Tuning as Control, Not a Flaw: The manual sweep of α₁ and α₂ is not a limitation but a core feature of our Pareto optimization framework. The goal is not to have the model magically find a single "best" trade-off. Different applications have different constraints; a latency-critical chatbot requires a different point on the Pareto frontier than a high-accuracy offline analysis tool. Our framework provides a principled control mechanism. By varying the α coefficients, a practitioner can explicitly select their desired cost-performance profile from a spectrum of pre-trained models. This is far more practical than re-training a model from scratch to meet new deployment needs. The expensive sweep is a one-time characterization of the frontier; deployment involves simply choosing a point on it.
>
>
> Weakness 3: SLM as a Symptomatic Fix (a "Patch").
>
> This is a very insightful critique. Your point that a model should learn to use CoT 100% on a math dataset is correct, if that were its only task. However, our goal is to build a general-purpose, adaptive assistant that retains its versatile capabilities across multiple stages of training.
>
> From this perspective, "decision boundary collapse" is a form of catastrophic forgetting. When fine-tuning on a specialized domain (like math), we want the model to get better at math reasoning, not to forget the general, adaptive triggering policy it learned on broad data. SLM is not a "patch" but a targeted mechanism to enable stable multi-stage learning. It "freezes" the triggering policy by masking its loss, allowing the PPO updates to focus solely on improving the quality of the reasoning tokens. This preserves the essential generalist skill (adaptive triggering) while allowing the specialist skill (math reasoning) to improve. It's a technique to manage the stability-plasticity dilemma inherent in multi-stage RL, and the stark failure of the no-SLM baseline in Table 1 demonstrates it is solving a real and critical problem, not just patching a flawed design.
>
> Weakness 4: Crucial Missing Experimental Baselines.
>
> This is the most significant and fair critique of our paper, and we fully agree. A claim of Pareto optimality is only meaningful when compared to competing methods. We apologize for this omission.
> Our initial experiments focused on ablations to demonstrate that our proposed components (Adaptive SFT, RL, SLM) were all necessary to construct the Pareto frontier shown in Figure 1. Figure 1 should thus be interpreted as a study of our method's internal components, establishing the trade-off space that AdaCoT itself can achieve.
>
> We failed to include external baselines that attempt to solve similar problems. This makes it impossible to judge whether our frontier is superior to others. We will significantly tone down claims like "pioneering adaptive triggering" in the Abstract and Introduction.
>
> Minor Issue: The paper title is incorrect.
> We are deeply sorry for this unprofessional mistake. We apologize for the poor impression this created.

---

> ### Author Response · Authors · 2025-12-04
> **Response to The Official Review (2/2)**
>
> Response to Questions
>
> Q1: Is SLM truly necessary if the reward structure was more robust?
>
> This is an excellent question. Simply removing the α₂ penalty during the RL-Math stage is unlikely to work. The model would indeed learn to use CoT for math, but its triggering policy would likely collapse to a simple "always on" state for any remotely complex prompt. When α₂ is re-introduced in a later general stage, the model would have to re-learn its nuanced, SFT-derived policy from scratch, which may not be possible or may lead to an unstable, oscillating policy. SLM's value is in preservation. It protects the finely-tuned, data-driven decision boundary learned in SFT, preventing it from being overwritten by a simple, biased policy learned during a specialized RL stage. It's about preserving a complex skill, not just fighting a reward signal.
>
> Q2: How sensitive is the framework to SFT data and how to deploy in a new domain?
>
> This is a critical practical question.
> Sensitivity: The initial SFT data is indeed crucial, as it provides the foundational "knowledge" of when CoT is beneficial. The 67% ratio in our SFT data reflects the distribution of our broad-domain training corpus. The framework is designed to be robust to this: the subsequent RL stage fine-tunes and adjusts this initial boundary based on the reward penalties.
> Deployment in a New Domain: A practitioner would not need to re-run the entire expensive α-coefficient sweep. The workflow would be:
> a. Start with one of our pre-trained AdaCoT models from the Pareto frontier (e.g., the "balanced" Exp3). This model already possesses a general adaptive capability.
> b. Fine-tune this model on the new domain data.
> c. If the new domain data is heavily biased (e.g., all legal queries require CoT), the practitioner should apply SLM during this fine-tuning to improve in-domain reasoning quality without destroying the model's general adaptiveness for out-of-domain queries.
> This makes deployment far more practical and modular.
>
>
> Once again, thank you for your incredibly valuable and challenging review. It has been instrumental in helping us see the weaknesses in our paper's positioning and has given us a clear path to significantly strengthen it.

---

### Official Review · Reviewer_Pnuk · 2025-10-29

**Soundness:** 3
**Presentation:** 2
**Contribution:** 2
**Rating:** 4
**Confidence:** 3

**Summary:**

The paper proposes AdaCoT which addresses the inefficiency of always using chain-of-thought (CoT) reasoning by enabling a model to adaptively decide when to invoke CoT. The paper frames this as a Pareto optimization problem balancing accuracy and cost, and addresses it using RL with rewards weighted by different coefficients. AdaCoT achieves significant token savings without substantially sacrificing performance.

**Strengths:**

1.  Addresses a Practical Efficiency Problem. AdaCoT directly tackles selective reasoning, reducing unnecessary CoT usage, which is highly relevant for large-scale or latency-sensitive applications.

2.  Selective Loss Masking (SLM) Innovation. Introduces SLM to stabilize RL training, preventing the trigger policy from collapsing to always CoT or always skip CoT. Ensures robust learning of the binary decision.

3.  Empirical Validation. Tested on both real-world production traffic and reasoning benchmarks.

**Weaknesses:**

1. The "Pareto optimization problem" is realized only through linear scalarization rather than by using an explicit algorithm to compute the Pareto front.

2. The label is derived from manually designed principles and relies on the capabilities identified by the internal LLM.

**Questions:**

1. For lines 233-235, could you clarify how the CoT necessity label is manually determined?

2. When exploring the 1D Pareto front, equation (5) involves two free parameters, $\alpha_1$ and $\alpha_2$. How should these two parameters be chosen to effectively cover the entire 1D Pareto front?

3. The authors argue that fully autonomous CoT trigger learning suffers from difficulty in measuring the counterfactual benefits of reasoning. Could this issue be mitigated by running the model twice: once with CoT and once without, and calculate the performance difference. That would double the running times of model, however it increases the reward accuracy.

4. Minor issue: the paper title is not updated.

---

> ### Author Response · Authors · 2025-12-04
> **Response to The Official Review (1/2)**
>
> Dear Reviewer,
>
> Thank you for your valuable time and insightful feedback.
>
> Below are our point-by-point responses to your weaknesses and questions.
>
> Response to Weaknesses and Questions
>
> Weakness/Question 1: "Pareto optimization" is realized only through linear scalarization. / How to choose α₁ and α₂ to cover the Pareto front?
>
> Thank you for raising this important point. You are correct that we use linear scalarization, which is a standard and widely-used technique for finding points on the Pareto frontier in multi-objective optimization.
>
> Why Linear Scalarization: This method is effective and computationally efficient for our problem. By treating the problem as max {P(θ) ， -T(θ)} (Equation 3), different weightings allow us to discover different optimal trade-offs. Our reward function (Equation 5) is a practical implementation of this principle, where adjusting α₁ and α₂ implicitly changes the weights (λP, λT) on performance versus cost. While more complex algorithms for computing the full Pareto front exist, linear scalarization provides a direct and controllable mechanism to generate diverse, Pareto-optimal policies within the established RLHF framework, which was our primary goal.
>
> Choosing α₁ and α₂: In our framework, these two coefficients control the penalties for two types of errors:
>
> α₁: Penalty for missing a required CoT (reasoning omission).
> α₂: Penalty for overusing CoT (reasoning overuse).
> To effectively explore the Pareto front, we control the ratio and magnitude of these penalties:
>
> To find models with lower CoT rates (prioritizing efficiency), one should set α₂ > α₁. This makes the model more conservative about using CoT. This is the strategy used in Exp1 (α₁=0.1, α₂=0.3).
> To find models with higher CoT rates (prioritizing performance), one should set α₁ > α₂. This encourages the model to use CoT more liberally to avoid the higher penalty for omission. This is the strategy used in Exp4 (α₁=0.3, α₂=0.1).
> Models like Exp2 and Exp3 represent intermediate points by adjusting these values.
> By systematically sweeping the ratio of α₁ to α₂, we can trace different points along the frontier. We have found this method to be intuitive and effective for generating a range of models with different performance-cost profiles, as demonstrated in Figure 1.
>
>
> Weakness/Question 2: Label dependency on manual principles and an internal LLM. / How is the CoT necessity label manually determined?
>
> Thank you for asking for more detail on this critical process. We apologize that the term "manually verified" may have caused confusion.
>
> Clarification of the Labeling Process: The initial data labeling is primarily automated. We do not manually label thousands of data points. Instead, we provide an auxiliary LLM (which shares the same architecture as our base model) with a set of explicit, high-level principles, which are detailed in appendix. These principles guide the LLM to self-assess the complexity of a query and decide whether CoT is necessary. For example, it checks if a query "requires multi-step reasoning" versus being answerable via "direct recall." This principle-guided, automated process is far more scalable and consistent than large-scale manual annotation.
>
> The Role of Manual Verification: The "manual verification" mentioned in Section 3.1 refers only to a final quality check on the 1000-prompt daily-use test set, which is used for evaluating the triggering accuracy (Table 1). This small, high-quality test set was manually reviewed to ensure its labels were accurate, making it a reliable benchmark for the model's decision-making capability. The large-scale SFT and RL training datasets rely on the automated, principle-guided labeling.

---

> ### Author Response · Authors · 2025-12-04
> **Response to The Official Review (2/2)**
>
> Question 3: Mitigating the counterfactual evaluation issue by running the model twice.
>
> This is an excellent suggestion and a valid alternative strategy for reward calculation. We did consider this approach during our design phase.
>
> Your proposed method, where one would run the model with and without CoT to calculate the performance difference as a reward signal, would indeed provide a more accurate, instance-specific reward for the CoT decision.
> However, we chose our current approach for two main reasons:
> - 1. Training Efficiency: As you correctly noted, this approach would double the computational cost of the RL training loop, as it requires two forward passes (one with CoT, one without) for every sample in a batch to calculate the reward. Given the massive scale of RLHF training, this overhead was a significant consideration.
> - 2. Reward Model Generalization: Our approach relies on a pre-trained reward model and CoT labels derived from a high-capability LLM. This is in line with standard RLHF practice. The core idea is that the reward model learns a general sense of response quality, and the penalties derived from the high-quality SFT data provide a strong, albeit imperfect, signal for CoT necessity. We found this combination to be a highly effective and efficient proxy for the true counterfactual benefit.
> While your proposed method offers higher reward accuracy, our method presents a more pragmatic and scalable trade-off for large-scale training.
>
> Minor Issue: The paper title is not updated.
>
> We are very sorry for this unprofessional error. This was an artifact from using the conference submission template, and it will be corrected in our revised version. We sincerely apologize for this oversight.
>
> Once again, we thank you for your sharp and constructive feedback. Your questions have helped us identify areas where we can improve the clarity and justification of our methodology. We hope our responses and the planned revisions have addressed your concerns effectively.

---

### Official Review · Reviewer_en53 · 2025-10-30

**Soundness:** 3
**Presentation:** 2
**Contribution:** 3
**Rating:** 6
**Confidence:** 3

**Summary:**

The paper tackles the practical question of when to trigger Chain-of-Thought (CoT) at inference time to balance quality and cost. It formulates the problem as Pareto optimization, maximizing task accuracy while minimizing the CoT triggering rate, and optimizes a linearly-scalarized objective with PPO, using asymmetric penalties for under-using and over-using CoT. To prevent decision-boundary collapse during multi-stage RL, the authors propose Selective Loss Masking (SLM) that masks the loss on the “decision token” right after `<think>` in fragile stages, preserving a healthy triggering distribution. Experiments across many public benchmarks and production traffic show strong accuracy–cost trade-offs (clear Pareto curves), significant token savings in the wild, and no loss of peak accuracy under an “always reason” setting.

**Strengths:**

This work focuses on the practical and important issue of adaptive Chain-of-Thought (CoT) triggering under tight efficiency constraints, where current systems tend to over-use CoT (wasting tokens/latency) or under-use it (hurting accuracy), and multi-stage RL can collapse the decision boundary to “always/never reason,” skewing usage and degrading robustness.

This work proposes a tightly coupled framework for Pareto-optimal adaptive CoT, combining a scalarized performance–cost objective with asymmetric penalties (to regulate under/over reasoning), PPO-based policy learning (to sweep the frontier), and Selective Loss Masking (SLM) (to stabilize multi-stage training).

This work conducts multiple sets of experiments to verify the effectiveness and extensibility of the proposed method.

**Weaknesses:**

(i) The contribution of this work reads primarily as a careful systems integration tailored to CoT triggering rather than a new theoretical principle, and the separation from adjacent paradigms, such as selector/gating approaches, early-exit/anytime inference, or logit/entropy-based stabilizers, remains somewhat blurred, making it hard to pinpoint what portion of the gains stems from genuinely new ideas.

(ii) The ablation does not include a systematic sensitivity analysis of key hyperparameters that govern the accuracy–cost trade-off and training stability (e.g., scalarization weights, penalty coefficients, and masking schedule). As a result, the robustness of the reported gains and the controllability of the Pareto frontier across seeds, datasets, and domains remain unclear, and it is difficult to disentangle performance attributable to the method from potential tuning artifacts.

(iii) In Related Work, this paper does not analyze in detail how the method addresses concrete failure modes of existing approaches (e.g., over-/under-triggering of CoT or decision-boundary collapse), nor does it map individual components (Pareto scalarization, asymmetric penalties, SLM) to these gaps.  This obscures the source of improvements and blurs the conceptual distinction from adjacent lines such as selector/gating, early-exit/anytime inference, and reasoning-length compression.

(iv) Has some loopholes, e.g.,

  - Format error, e.g., The title of this article uses the sample provided by the template and has not been modified to the title of this article.

Please correct the grammatical mistakes and polish them if possible.

**Questions:**

Please see 'weakness', which simply can be summarised as:

(i) Please articulate the core conceptual novelty beyond linear scalarization with PPO and the SLM masking trick, and clarify which portion of the gains stems from genuinely new ideas.

(ii) Please provide a systematic sensitivity analysis of the key hyperparameters that control the accuracy–cost trade-off and training stability (e.g., ($\lambda_P$, $\lambda_T$, $\alpha_1$, $\alpha_2$) and the SLM masking schedule), including variance across random seeds/datasets and the resulting changes to the Pareto frontier.

(iii) Please expand the Related Work (or an analysis section) to explicitly map each component—Pareto scalarization, asymmetric penalties, and SLM—to concrete failure modes in prior approaches (e.g., over-/under-triggering of CoT, decision-boundary collapse), and please substantiate these links with targeted evidence or case studies.

(iv) Please fix the presentation/formatting issues and please polish the writing: update any placeholder elements (e.g., the template title).

---

> ### Author Response · Authors · 2025-12-04
> **Response to The Official Review (1/2)**
>
> Dear Reviewer,
>
> We sincerely thank you for your time and for providing such a detailed and supportive review of our work.
> Below are our point-by-point responses to your weaknesses.
>
> Response to Weaknesses
>
> (i) The core conceptual novelty beyond system integration.
>
> Thank you for this crucial question, which prompts us to articulate our conceptual contributions more sharply. While AdaCoT is indeed a carefully integrated system, its novelty lies in two key areas:
>
> A New Formalization for a Key Problem: Our primary conceptual contribution is framing the adaptive CoT triggering problem as a Pareto optimization task. Prior works often relied on heuristic methods or focused on a different problem (reasoning length compression). By formalizing the trade-off between performance and CoT usage , we provide a principled, mathematical foundation to tackle this challenge. This formalism naturally leads to an RL-based control mechanism where the policy can be steered along the Pareto frontier. This is conceptually distinct from selector/gating approaches that typically learn a single, fixed classification boundary.
>
> Identifying and Solving a Critical RL Instability: Our second key contribution is the identification and solution to "decision boundary collapse" in multi-stage RL for this task. This is a real and significant practical problem we encountered, where a well-trained adaptive policy is easily destroyed by subsequent fine-tuning on domain-specific, biased data (e.g., math datasets). Selective Loss Masking (SLM) is our novel and specific solution to this instability. While related to general stabilization techniques, SLM is a targeted intervention, surgically masking the loss only on the pivotal "decision token" to preserve the learned triggering distribution. The ablation study in Table 1, showing the dramatic failure of RL-Math without SLM (Accuracy collapses from 0.813 to 0.506), provides direct evidence of both the problem's existence and our solution's efficacy.
>
> Therefore, the gains stem from this combination: the Pareto formalism provides the "what" (the optimization goal), and SLM provides the "how" (the stability needed to achieve it via multi-stage RL).
>
> (ii) The lack of a systematic sensitivity analysis of key hyperparameters.
>
> This is a very fair point. A thorough sensitivity analysis is crucial for understanding the robustness and controllability of our method. While our main experiments (Exp1-Exp4) demonstrate that varying α₁ and α₂ effectively steers the model along the Pareto frontier, we agree that a more systematic analysis is needed.
>
> Due to the rebuttal's time and computational constraints, conducting a full-scale analysis across all seeds and datasets is challenging. However, we can provide further insights based on our internal development experiments and commit to adding a more detailed analysis in the appendix of the final version.
> Our observations are:
> The ratio of α₁ (penalty for missing CoT) to α₂ (penalty for overusing CoT) is the primary driver of the final CoT triggering rate. Increasing α₁ or decreasing α₂ reliably pushes the model to trigger CoT more often, as shown by the progression from Exp1 to Exp4.
> The γ (format error penalty) is a stability hyperparameter. Setting it to a high value (e.g., 1.0) was effective in quickly teaching the model the desired <think></think> format, and its exact value was not highly sensitive once it was sufficiently large.
> SLM is a binary choice (on/off) rather than a tunable hyperparameter. We found it essential for the RL-Math stage but unnecessary for the RL-General stage, which used a more balanced dataset.

---

> ### Author Response · Authors · 2025-12-04
> **Response to The Official Review (2/2)**
>
> (iii) In Related Work, a clearer mapping of components to gaps in prior work is needed.
>
> Thank you for this excellent suggestion. We agree that explicitly connecting our components to the failure modes of existing approaches would significantly strengthen our paper's narrative and clarify our contributions.
>
> Prior approaches can be broadly categorized:
> Monotonic Reduction Methods (e.g., length compression, brevity rewards): These methods fail to account for query complexity, often indiscriminately shortening reasoning for all inputs. This risks under-triggering or providing insufficient reasoning for complex queries where it is needed. Our Pareto scalarization with asymmetric penalties directly addresses this by learning a selective triggering policy that balances cost and the performance gain from reasoning.
> Fixed Selectors/Gating Models: These methods typically learn a single, static policy for when to use CoT. They are brittle and suffer from what we identify as "decision-boundary collapse" when subjected to further fine-tuning on specialized data, a common practice in LLM development. Our Selective Loss Masking (SLM) is the specific technical contribution designed to solve this exact failure mode, ensuring the adaptive policy remains robust across training stages.
> Manual/Heuristic Mechanisms: These approaches lack a principled optimization framework and are not learned end-to-end, making them less robust and generalizable. Our entire RL-based framework provides this principled, learned approach.
>
>
> (iv) Addressing formatting and grammatical issues.
>
> We sincerely apologize for the unprofessional formatting errors and any grammatical mistakes. These were oversights during the preparation of the submission draft and are not reflective of the care we have taken with the research itself.
>
> Once again, we express our profound gratitude for your valuable and supportive review. Your feedback has provided us with a clear roadmap to substantially improve the paper's clarity and impact. We hope our responses have addressed your points satisfactorily.

---

### Official Review · Reviewer_xvjU · 2025-11-03

**Soundness:** 2
**Presentation:** 2
**Contribution:** 3
**Rating:** 4
**Confidence:** 2

**Summary:**

This paper proposes adaptive chain-of-thought (CoT), in which large language models (LLMs) adaptively decide whether to invoke CoT reasoning. The intuition is that CoT is often time- and resource-consuming, and that it is often not needed for simple queries. AdaCoT is formulated as a Pareto optimization problem of maximizing accuracy while minimizing computational overhead, and the paper proposes to use proximal policy optimization (PPO) to allow LLMs to decide whether to invoke CoT on a given query. In training the PPO algorithm, the paper employs selective loss masking to ensure robustness, and experiments demonstrate that the learned PPO algorithm can successfully reduce the number of CoT invocations without degrading performance on complex tasks.

**Strengths:**

+ The paper proposes an interesting idea that makes intuitive sense. The experiments (Figure 1) also seem to validate that a CoT triggering rate of around 60% achieves essentially the same average score as always invoking CoT.

+ The experiments include evaluation on fifteen different benchmark datasets, although Table 4 reporting the results on these datasets is borderline unreadable due to the small font used. AdaCoT has substantially lower triggering rates but comparable performance to full CoT, with better scores than reinforcement learning baselines that do not use CoT.

**Weaknesses:**

--AdaCoT’s training process requires labels of whether to use CoT on a set of queries, which the paper proposes to obtain from from another model. Why not use this model as the discriminator for whether to invoke CoT? If invoking this model is too expensive, why not use a form of knowledge distillation to learn a smaller discriminator model instead of reinforcement learning?

--It’s not clear why reinforcement learning is used to decide whether to invoke CoT. Reinforcement learning is typically useful when there are meaningful transition dynamics or reward coupling between states, but in this case the transitions seem to be determined by the incoming queries, and there is no reward coupling. Moreover, the reinforcement learning framework is under-specified. What are the actions and rewards? How often is the selective loss masking used; is it used in every training iteration (in which case, why have the decision token loss at all)?

--The paper does not include a comparison with models that aim to optimize reasoning length, even though such models would also likely reduce inference time and computational overhead. Wouldn’t a reasoning length of 0 in such models be equivalent to not using CoT, and thus wouldn’t such models be a generalization of the proposed AdaCoT framework?

**Questions:**

1) In the first sentence of Section 5.1, two “internal…mixture-of-experts” models are introduced as the “base model”s. Are these the LLMs using CoT, or the models generating the CoT labels? What does “internal” mean?

2) Figure 1 shows a fairly small range of CoT invocation rates. Thus, it’s not clear that the proposed weighting of different reinforcement learning rewards will really allow AdaCoT to traverse the Pareto front.

3) Figure 2 is discussed in the first paragraph of Section 4.2, but I could not find a second figure anywhere in the paper’s main body or appendix.

4) This is not really a question, but the paper title is not included in the draft (which has the title “Formatting Instructions for ICLR 2026 Conference Submissions”).

Please see also “Weaknesses” above.

---

> ### Author Response · Authors · 2025-12-04
> **Response to The Official Review  (1/2)**
>
> Dear Reviewer,
>
> We sincerely thank you for your time and for providing such insightful and constructive feedback on our manuscript.
>
> Below are our point-by-point responses to your weaknesses and questions.
>
> Response to Weaknesses
>
> Weakness 1: Why not use the labeling model as a discriminator or use knowledge distillation (KD)?
>
> Thank you for this insightful question regarding the choice of our training methodology over simpler alternatives. We apologize for the lack of clarity in our description, which we have now rectified in the revised manuscript.
>
> Clarification on the Labeling Model: The "auxiliary model" used for initial data labeling is, in fact, a fine-tuned version of the very same pre-trained base model that we use for AdaCoT. We described it as "another model" for simplicity, but your question highlights that this phrasing was misleading. The labeling model first generates a reasoning process and then self-assesses whether CoT was necessary, providing a high-quality, capability-aligned label. Using a model with a similar capability frontier for labeling is crucial for accuracy.
>
> Why Reinforcement Learning over a Static Discriminator/KD: The goal of AdaCoT is not just to learn a single, fixed decision boundary, but to create a framework for exploring the entire Pareto frontier of performance versus cost.
>
> A static discriminator (either the labeling model itself or a distilled version) would learn to replicate a single, predetermined CoT triggering strategy from the SFT data.
> Our RL-based approach, however, provides controllability. By adjusting the penalty coefficients in the reward function during the RLHF stage, we can steer the model's policy to different decision thresholds. This allows us to train multiple models (like Exp1-Exp4) that land on different points of the Pareto curve from a single SFT-initialized model and a single RL dataset. This flexibility is a core contribution of our framework that a static discriminator or KD model would not provide.
>
>
> Weakness 2: Why Reinforcement Learning? Under-specified RL framework and Selective Loss Masking (SLM).
>
> We thank the reviewer for pointing out the need for greater clarity regarding our RL framework.
>
> Rationale for RL: Our use of "Reinforcement Learning" refers to the standard post-training pipeline for modern LLMs, commonly known as RLHF (Reinforcement Learning from Human Feedback), where PPO is used to optimize the language model's policy. Our novelty lies in how we adapt this established process for a new goal.
>
> Clarification on Selective Loss Masking (SLM): We apologize for the insufficient explanation. The reviewer's intuition is correct—if SLM were used everywhere, the decision token's loss would be meaningless. SLM is used selectively and only in specific, challenging training stages.
>
> As stated in Section 3.1, we apply SLM only during the Mathematics-Focused RL stage (RL-Math). This stage uses a dataset where virtually every prompt requires CoT.
> Without SLM, any model rollout that receives positive reward in this stage must have triggered CoT. The PPO updates would therefore overwhelmingly reinforce the "always trigger CoT" behavior, causing the nuanced, adaptive capability learned during SFT to collapse.
> SLM elegantly prevents this "decision boundary collapse" by masking the loss from the decision token, thereby protecting the learned triggering policy. This allows the RL process to focus on improving the quality of the reasoning steps themselves, without corrupting the adaptive trigger. It is a crucial technique for stable multi-stage RL training on biased data distributions.
>
>
> Weakness 3: Comparison with reasoning length optimization models.
>
> This is an excellent point. We agree that methods optimizing for reasoning length are highly relevant and also target computational efficiency.The core distinction is that AdaCoT addresses a different, albeit related, question.
>
> Reasoning Length Optimization primarily asks: "Given that we are reasoning, how can we make the reasoning path shorter and more concise?" A reasoning length of 0 is conceptually equivalent to our "No CoT" case.
> AdaCoT asks the prerequisite question: "Is reasoning necessary at all for this query?"
> Our framework focuses on learning the binary decision to invoke CoT, which we believe is a fundamental and currently under-explored aspect of efficient reasoning. While many length optimization works are tailored to specific tasks with verifiers (e.g., math problems), our method is designed to be general and integrates smoothly into a standard RLHF pipeline that uses a general-purpose reward model, making it broadly applicable to both rule-verifiable and more subjective, general-domain tasks.
>
> We acknowledge this is a valuable comparison. In our revised manuscript, we will expand the Related Work section to more directly discuss the relationship and distinction between our adaptive triggering framework and reasoning length compression techniques.

---

> ### Author Response · Authors · 2025-12-04
> **Response to The Official Review (2/2)**
>
> Response to Questions
>
> Q: Clarification on "internal...mixture-of-experts" models.
> We sincerely apologize for the confusing phrasing. "Internal" refers to proprietary models developed within our organization, as opposed to open-source models. The phrase "15B/150B parameter Mixture-of-Experts (MoE) model" refers to a single MoE model, not two. This model has a total of 150B parameters, of which a sparse mixture of 15B parameters are active for any given input. This is the base model used for all experiments in the paper. We have corrected this phrasing to avoid ambiguity.
>
> Q: The range of CoT invocation rates in Figure 1 seems small.
> This is a very keen observation. The range shown for our AdaCoT RL models (from 43% to 68%) represents the most critical "region of interest" on the Pareto frontier, where performance is highly competitive with the Full CoT baseline. The extremes of the frontier are already represented by our No CoT RL (0% rate) and Full CoT RL (100% rate) baselines. Our experiments were designed to explore this high-performance trade-off zone. Pushing the CoT rate much lower would cause performance to drop towards the No CoT baseline, which is less interesting for practical applications. The specific range is also influenced by the nature of the 15 selected benchmarks, which contain many complex reasoning tasks, making a moderate-to-high CoT rate necessary to maintain high performance. We will add a sentence in our analysis of Figure 1 (Section 3.2.1) to clarify this point.
>
> Q: Figure 2 is discussed but could not be found.
> Our sincerest apologies for the confusion. Figure 2, which illustrates the meta-reasoning response structure, is indeed present in the manuscript. It is located on page 22 in the appendix.
>
> Q: The paper title is incorrect in the draft.
> We are very sorry for this unprofessional error. This was an artifact from using the conference submission template, and it will be corrected in our final version. We sincerely appreciate you pointing this out and apologize for the poor impression it may have given.
>
> Once again, we thank you for your thoughtful and detailed review. Your feedback has been invaluable in helping us improve the clarity and impact of our paper.

---

### Meta-Review · Area_Chair_Yroj · 2026-01-04

**Summary:**

The reviewers raised the following key concerns. First, the paper claimed Pareto optimal results but failed to provide a comprehensive comparison against other efficient reasoning methods beyond simple baselines. Second, the current submission didn't provide a systematic sensitivity study of key hyperparameters. Third, reviewer moGN noted the limitation of the binary decision formulation. Finally, there were concerns regarding clarity and paper presentation.

**Reviewer Concerns:**

The rebuttal largely addressed concerns regarding paper presentation and answered most clarification questions. It also partially addressed the concerns regarding the binary decision formulation. However, due to time constraints, the rebuttal didn't include comparisons against other baselines or a systematic study on hyperparameters, leaving the first two concerns outstanding.

**Reviewer Scores:**

If the reviewers had been able to participate fully in the discussion, I believe reviewer moGN might have increased their score from 2 to 4, while other reviewers' scores would likely have remained unchanged.

---

### Decision · Program_Chairs · 2026-01-26

Reject